Predicting the contribution of climate change on North Atlantic underwater sound propagation

Possenti Luca 1 luca.possenti@nioz.nl
http://orcid.org/0000-0002-7256-2243 Reichart Gert-Jan 1 2
de Nooijer Lennart 1
Lam Frans-Peter 3
de Jong Christ 3
Colin Mathieu 3
Binnerts Bas 3
Boot Amber 4
http://orcid.org/0000-0002-5557-3282 von der Heydt Anna 4 5
1 Ocean Systems (OCS), Royal Netherlands Institute for Sea Research (NIOZ) , Texel , The Netherlands
2 Department of Earth Sciences—Faculty of Geosciences, Utrecht University , Utrecht , The Netherlands
3 Acoustics & Sonar, Netherlands Organization for Applied Scientific Research (TNO) , The Hague , The Netherlands
4 Department of Physics, Institute for Marine and Atmospheric research Utrecht (IMAU)—Faculty of Science, Utrecht University , Utrecht , The Netherlands
5 Centre for Complex Systems Studies, Utrecht University , Utrecht , The Netherlands
Fu Guobin
Electronic publication date: 2023 Oct 10
Publication date: 2023
Volume: 11
Electronic Location ID: e16208
Received 2023 Jun 19; Accepted 2023 Sep 8
Copyright: © 2023 Possenti et al.
Copyright year: 2023
Copyright holder: Possenti et al.
License: This is an open access article distributed under the terms of the Creative Commons Attribution License, which permits unrestricted use, distribution, reproduction and adaptation in any medium and for any purpose provided that it is properly attributed. For attribution, the original author(s), title, publication source (PeerJ) and either DOI or URL of the article must be cited.
License URL: https://creativecommons.org/licenses/by/4.0/

Keywords: Climate change, North Atlantic Ocean, Underwater acoustics, Shipping noise, Human impact, Sound propagation, Atlantic Meridional Overturning Circulation

Funding: SOUND-2 Project P17-07 Research Program AQUA Dutch Research Council (NWO) Program of the Netherlands Earth System Science Centre (NESSC) Ministry of Education, Culture and Science 024.002.001 This publication is part of the SOUND-2 project with project number P17-07 of the research program AQUA which is financed by the Dutch Research Council (NWO). Anna von der Heydt and Amber Boot were supported under the program of the Netherlands Earth System Science Centre (NESSC), financially supported by the Ministry of Education, Culture and Science (OCW, Grantnr. 024.002.001). The funders had no role in study design, data collection and analysis, decision to publish, or preparation of the manuscript.

==============================
Since the industrial revolution, oceans have become substantially noisier. The noise increase is mainly caused by increased shipping, resource exploration, and infrastructure development affecting marine life at multiple levels, including behavior and physiology. Together with increasing anthropogenic noise, climate change is altering the thermal structure of the oceans, which in turn might affect noise propagation. During this century, we are witnessing an increase in seawater temperature and a decrease in ocean pH. Ocean acidification will decrease sound absorption at low frequencies (<10 kHz), enhancing long-range sound propagation. At the same time, temperature changes can modify the sound speed profile, leading to the creation or disappearance of sound ducts in which sound can propagate over large distances. The worldwide effect of climate change was explored for the winter and summer seasons using the (2018 to 2022) and (2094 to 2098, projected) atmospheric and seawater temperature, salinity, pH and wind speed as input. Using numerical modelling, we here explore the impact of climate change on underwater sound propagation. The future climate variables were taken from a Community Earth System Model v2 (CESM2) simulations forced under the concentration-driven SSP2-4.5 and SSP5-8.5 scenarios. The sound modeling results show, for future climate change scenarios, a global increase of sound speed at different depths (5, 125, 300, and 640 m) except for the North Atlantic Ocean and the Norwegian Sea, where in the upper 125 m sound speed will decrease by as much as 40 m s−1. This decrease in sound speed results in a new sub-surface duct in the upper 200 m of the water column allowing ship noise to propagate over large distances (>500 km). In the case of the Northeast Atlantic Ocean, this sub-surface duct will only be present during winter, leading to similar total mean square pressure level (SPLtot) values in the summer for both (2018 to 2022) and (2094 to 2098). We observed a strong and similar correlation for the two climate change scenarios, with an increase of the top 200 m SPLtot and a slowdown of Atlantic Meridional Overturning Circulation (AMOC) leading to an increase of SPLtot at the end of the century by 7 dB.

Introduction

The natural soundscape is altered by anthropogenic activities such as shipping, transport, oil and gas exploitation, defense activities, tourism, fishing, offshore wind farming, and on- and near-shore construction (Richardson et al., 2013; Duarte et al., 2021). Among these the main anthropogenic noise source in the oceans is shipping, which dominates the soundscape in the low-frequency range (10 Hz to 1 kHz) (Wenz, 1962). Under 300 Hz this effect increased in the past 50–60 years because regions exposed to intense ship traffic have experienced an increase in ambient noise. In these regions ambient noise increased by 3 dB decade−1 (Andrew et al., 2002; Andrew, Howe & Mercer, 2011; Chapman & Price, 2011; Erbe et al., 2019; Miksis-Olds, Bradley & Maggie Niu, 2013; Miksis-Olds & Nichols, 2016), resulting in an absolute sound increase by 15 to 20 dB (Andrew et al., 2002; McDonald, Hildebrand & Wiggins, 2006; McKenna et al., 2012). A major component of this increase is given by the rise in the number of ships, which is estimated to have doubled in the period between 1965 to 2000 (from approximately 44,000 to 88,000) (Hildebrand, 2009). Future estimates suggest that with the current rate of growth in ship traffic and economic trading, ambient noise is projected to continue to rise globally, especially in the Arctic and around Africa (United Nations, 2021). At the same time, humankind has introduced more than 330 Petagram of CO2 into the atmosphere since the industrial revolution (starting around 1,760) (Canadell et al., 2007). A substantial part of the added CO2 has been absorbed by the ocean (about 25%, (Watson et al., 2020)), which affected the oceanic carbon system. Global average surface ocean pH decreased from 8.21 to 8.1, corresponding to a 29% increase in H+ activity (Doney et al., 2009; Fabry et al., 2008). Future projections suggest that in the next decades ocean CO2 uptake will continue, decreasing the ocean pH in a process known as Ocean Acidification (OA). OA is adversely affecting the ocean environment by lowering sound absorption (α) at frequencies below 10 kHz which is controlled by pH-dependent borate ion chemistry (Francois & Garrison, 1982). At higher frequencies (>10 kHz) absorption is not affected by pH because this mechanism depends primarily on the chemical relaxation of magnesium sulfate and pure water viscous absorption. Largest relative pH-driven reduction in sound absorption will occur in the low frequency range, reaching values as high as a 40% reduction under 500 Hz (Hester et al., 2008; Brewer & Hester, 2009). Such a reduction in absorption will allow sound to travel further in situations when absorption is the dominant component in propagation loss (PL). The contribution of sound absorption at low frequencies (<500 Hz) is minimal, making its present and future contribution to the propagation loss negligible (Udovydchenkov et al., 2010; Reeder & Chiu, 2010; Joseph & Chiu, 2010). However, when sound is trapped in a channel or duct in the ocean that may form as a consequence of the ocean’s thermal structure and propagates over large distances, the latter effect may become important.

In addition to ongoing ocean acidification the ocean soundscape is primarily affected by other climate-related processes such as ocean warming, changes in wind speed and storm intensity and frequency, increase in sea-ice melting and decrease in salinity (Ainslie et al., 2021; Andrew et al., 2002; Duarte et al., 2021; Munk, 2011; Young, Zieger & Babanin, 2011). For the period between 1971 to 2020 the total heat system had a heat gain of 381 ± 61 ZJ with an associated total heating rate of 0.48 ± 0.1 W m−2 and about 89% of this heat is stored in the ocean (Von Schuckmann et al., 2023). The temperature increase over the entire profile, together with sea-ice melting, is projected to alter the ocean’s sound speed (c) profile. For a Representative Concentration Pathway RCP8.5, Affatati, Scaini & Salon (2022) quantified a general increase of sound speed up to 20 m s−1 (1.5%) at the end of the current century.

The impact of climate change on marine ecosystems has been widely researched but implications are largely unknown. These changes in sound propagation due to climate change may have a substantial effect on marine mammals with specialized auditory systems (Wartzok et al., 2004). Marine mammals use sound for various functions such as competition to show territorial hegemony, predation, mating and warning of others about presence of predators (Au & Hastings, 2008).

Potential impacts of climate change on the ocean’s soundscape have received relatively little attention even though they may affect biology profoundly. The last assessment by the IPCC of climate change impacts (Skea, Shukla & Kılkış, 2022) did not acknowledge climate change related impacts on the ocean soundscape, whereas the IPCC report on oceans and the cryosphere only acknowledged noise in the context of increased human operations in the Arctic Ocean (Poloczanska et al., 2018) related to sea-ice melting.

Here we investigate the correlation between climate-related changes and the future sound propagation. We predict the expected changes by the end of the century in the sound field produced by a single vessel at 125 Hz using two different climate scenarios: Shared Socioeconomic Pathways SSP5-8.5 and SSP2-4.5 (Riahi et al., 2017; O’Neill et al., 2014). We selected a 125 Hz source frequency because (together with 63 Hz) it is a frequency band specified by the European MFSD (Marine Strategy Framework Directive) to assess the changes in ambient noise (Van der Graaf et al., 2012). In a later study de Jong et al. (2021) recommended to use also higher frequencies to monitor ambient noise (e.g., 1 kHz). We contrast different areas globally to investigate spatial differences in the underwater sound propagation and compare different scenarios.

Materials and Methods

Climate change data

To calculate the future change in PL, we retrieved 1 year of three-dimensional (3-D) monthly mean fields of salinity (S), temperature (T), pH and two-dimensional (2-D) monthly mean fields of near-surface (2 m) air temperature, Atlantic Meridional Overturning Circulation (AMOC) stream function and wind speed from 2022 to 2099 from the Community Earth System Model v2 (CESM2; Danabasoglu et al., 2020) as simulated in the Coupled Model Intercomparison Project 6 (CMIP6; Eyring et al., 2016) using the concentration driven SSP2-4.5 (Danabasoglu, 2019) and SSP5-8.5 scenarios (Danabasoglu, 2019) from the simulation r11i1p1f1. The ocean circulation model in CESM2, POP2 (Smith et al., 2010), uses a nominal horizontal resolution of 1° (100 km) on a displaced Greenland grid with 60 non-equidistant layers in the vertical. To fill the T, S and pH depth gaps we interpolated the profiles using a shape-preserving piecewise cubic interpolation leading to the final profiles with 1 m resolution.

Calculation of propagation loss

To determine changes in the future sound propagation we calculated the sound field produced by a single typical merchant vessel. We used as input the monopole vessel Source Level (SL) of 170.1 dB re 1 μPa m using the model proposed by MacGillivray & de Jong (2021) at a single frequency (125 Hz) for a bulker with reference length and speed (211 m and 13.9 kn).

To analyze the PL we selected six locations globally, including the North Atlantic Ocean. Locations used are two in the Atlantic Ocean (45° N, 40° W and 48° N, 14° W), Pacific Ocean (50° N, 167° E), Southern Ocean (55° S, 140° E), Arctic Ocean (75° N, 140° W) and Norwegian Sea (72° N, 1° W).

In each location, we placed the sound source at 6 m depth and we calculated the PL using the RAM parabolic equation model (Collins, 1995). We used the ocean bathymetry available from the 2022 General Bathymetry Chart of the Oceans (https://www.gebco.net/data_and_products/gridded_bathymetry_data/). In the model, we evaluated the effect of climate change and ocean acidification using sound speed and potential density (σ0) mean profiles for boreal winter and summer (2018 to 2022) and (2094 to 2098).

We calculated sound speed and potential density using the equations of Roquet et al. (2015) that require temperature, salinity and pressure as inputs. Also, we calculated sound absorption using the formula of van Moll, Ainslie & van Vossen (2009) requiring an input of temperature, salinity, sound frequency (f), depth (z) and pH.

The final PL was calculated using a constant bathymetry from the sound source with a vertical resolution for PL of 1 m to a maximum distance of 500 km. Also, we assumed at every location the same sediment composition of very fine silt (grain size 8 μm), being the median grainsize for sediments in deep waters (Ainslie, 2010).

To calculate the sound pressure level of the ship (SPLship) as a function of range and depth, the PL has been subtracted from the SL and then added to the ambient noise sound pressure level (SPLwind) calculated using a composite wind model (Ainslie, 2010). To calculate SPLwind, we assumed a homogenous wind surface source factor calculated using seasonal average wind speed. Subsequently we calculated SPLwind as a function of depth propagating wind noise using a constant sound speed profile. The final result is presented as SPL over distance from the ship when SPLship (ship noise) exceeds SPLwind presented as the total mean square pressure (SPLtot) of the sum of SPLship and SPLwind. Also, when determining the effect of sound absorption, the difference of SPLtot in the two different runs (with and without sound absorption) was calculated.

Subsequently, we present SPLtot over 500 km for the entire water column for the six selected locations in the summer and winter seasons for SSP5-8.5 and SSP2-4.5 climate scenarios. In the results and discussion sections we discuss the differences between the six locations and the two seasons. To help the reader the methods applied in this section are synthesized in Table 1.

Table 1 Methods summary.

Letter name	Method	Input	
Source level (SL)	MacGillivray & de Jong (2021)	Vessel length: 211 m
Vessel speed 13.9 kn	
Propagation loss (PL)	RAM	Bathymetry GEBCO
Temperature profile CESM2
Salinity profile CESM2
pH profile CESM2	
Ambient noise (SPLwind)	Ainslie (2010)	Temperature profile CESM2
Air temperature CESM2
pH profile CESM2
Wind speed CESM2
Salinity profile CESM2	
Note:

Methods used to calculate the final sound pressure level (SPLtot) at 125 Hz for SSP5-8.5 and SSP2-4.5 scenarios.

Also, we relate the changes in SPLtot in the North Atlantic Ocean (45° N–40° W) with changes in AMOC strength for SSP2-4.5 and SSP5-8.5 smoothed using a 5 years moving median of the yearly mean SPLtot in the surface 200 m between 475 to 500 km distance from the source.

The final values of SPLtot aims at identifying regions with the largest impact of predicted climate change on sound propagation. In order to calculate the absolute changes in noise levels all ambient noise sources should ideally be incorporated and a more complete wind model would be required. However, we expect the impact of using a more complex wind model to be minimal, given the high uncertainty of the wind within the climate data.

Results

The trend in future sound speed

By 2098 sound speed will have increased (compared to 2022) globally under the SSP5-8.5 scenario at all depths below 300 m, with a predicted maximum increase in the North Atlantic Ocean, Labrador Sea and Norwegian Sea while it will decrease in the top 125 m (Fig. 1). In the less extreme SSP2-4.5 scenario the surface sound speed is also expected to increase, albeit less than under the SSP5-8.5 scenario with a maximum increase in the Arctic Ocean at 5 m below sea surface and the North Atlantic Ocean under 300 m (Fig. 2). In other regions such as the South Atlantic, Indian and Central Pacific Oceans the changes in sound speed will be negligible. For SSP2-4.5 scenario, the surface sound speed in the North Atlantic Ocean and Norwegian Sea is projected to decrease by >30 m s−1. In other locations such as the Pacific Ocean at around 0° N at 125 m and the Southern Ocean in the top 5 m sound speed is expected to remain more or less similar.

Figure 1 Sound speed difference between (2018 to 2022) and (2094 to 2098) for SSP5-8.5.

Maps of the difference in 5 years mean of sound speed (c) in m s−1 between (2018 to 2022) and (2094 to 2098) at (A) 5, (B) 125, (C) 300 and (D) 640 m depth calculated for SSP5-8.5. The black dots indicate the sound source locations.

Figure 2 Sound speed difference between (2018 to 2022) and (2094 to 2098) for SSP2-4.5.

Maps of the difference in 5 years mean of sound speed (c) in m s−1 between (2018 to 2022) and (2094 to 2098) at (A) 5, (B) 125, (C) 300 and (D) 640 m depth calculated for SSP2-4.5. The black dots indicate the sound source locations.

Sound speed profiles at the locations selected for the PL analysis for the winter and the summer seasons show a general sound speed increase in the top 2,000 m for SSP5-8.5 and SSP2-4.5 (Fig. 3). For SSP5-8.5 in the North Atlantic Ocean (47° N, 14° W) the sound speed will be similar in the top 100 m, increasing up to 40 m s−1 in the deeper water column. A smaller increase is observed at another location in the North Atlantic Ocean (45° N, 40° W) with 10 m s−1. The SSP2-4.5 shows the same overall trends, but with different values: for example at 47° N, 14° W the increase in sound speed is 40 m s−1 and at the surface at 45° N, 40° W it decreases by is 5 m s−1.

Figure 3 Sound speed profiles for the selected locations.

Sound speed (c) in m s−1 profiles over depth for the winter (dashed lines) and summer season (continuous lines) where in blue is boreal summer (2018 to 2022), in red winter (2018 to 2022), in yellow summer (2094 to 2098) and in purple winter (2094 to 2098) for SSP5-8.5 and green and azure for summer and winter (2094 to 2098) for SSP2-4.5 for (A) Northwest Atlantic Ocean (45° N 40° W), (B) Northeast Atlantic Ocean (47° N 14° W), (C) Norwegian Sea (72° N 1° W), (D) Arctic Ocean (75° N 140° W), (E) North Pacific Ocean (50° N 167° E) and (F) Southern Ocean (60° S 25° E).

Focusing on specific locations in the North Atlantic Ocean (Figs. 3A and 3B, 45° N, 40° W and 47° N, 14° W), in 2022 there are two sound speed minima present: at 150 m and from 150 to 500 m. At the end of the century, the surface minimum will be more marked and the deep sound channel will deepen to 1,500 m. The deep sound channel is located at the slowest sound speed where sound waves can travel long distances. The difference in sound speed from the sub-surface channel and the layer below in 2022 is 1 m s−1, which will increase to 24 and 20 m s−1 for SSP2-4.5 and SSP5.8.5, respectively. In 2022 the Norwegian Sea (72° N, 1° W) shows a surface sound speed minimum that in the SSP2-4.5 and SSP5-8.5 scenario will disappear in favor of a sub-surface duct at 100 m (Fig. 3C). The sub-surface duct is defined by the sound speed minimum in the top 500 m and sound can travel large distances because continually bent, or refracted, towards the region of lower sound speed. In both climate scenarios tested a deep sound channel will develop at 1,000 m before 2098.

In the Arctic Ocean (75° N, 140° W) in 2022, a surface sound speed minimum is present that will have deepened by 2098 to 85 m (Fig. 3D). In the more severe climate change scenario (SSP5-8.5) this change is more pronounced compared to the moderate scenario SSP2-4.5 (Fig. 3). During winter the difference in sound speed from the surface to 85 m will be 2 and 10 m s−1 for SSP2-4.5 and SSP5-8.5, respectively. In the North Pacific Ocean (50° N, 167° E), profiles in 2022 and 2098 are similar, with a sub-surface sound speed minimum located around 115 m depth (Fig. 3E). This minimum is expected to weaken over time, with a sound speed decrease in summer in the upper 115 m by 18 m s−1, which will decelerate to a decrease by 2098 of 12 m s–1 for SSP5-8.5 and SSP2-4.5. In the Southern Ocean (Fig. 2F, 60° S, 25° E), the surface sound speed will also increase by >10 m s−1 with an increase for SSP5-8.5 during summer (>15 m s−1). However, the absolute change of sound speed has a minor effect on sound propagation because it is not affecting stratification.

Trends in future sound absorption

The other analyzed variable potentially impacted by anthropogenic carbon addition and climate change is absorption. At the analyzed frequency of 125 Hz by the end of the century at 5, 125, 300 and 640 m depth, changes in sound absorption are small, with a minor decrease in absorption at 5 m around 80° N of no more than 0.0016 dB km−1 that is just 0.8 dB at 500 km from the source. The decrease in absorption will be smaller closer to the equator, with a small decrease of 0.0004 dB km−1 (0.2 dB at 500 km) at 0° N. Another aspect is that these changes will disappear over depth with values close to 0 dB km−1, for example from 640 m downward in the Pacific Ocean (Figs. 4 and 5). The decrease of absorption will be smaller in SSP2-4.5 compared to SSP5-8.5 where the largest decrease will be in the Arctic Ocean at 5 m with changes in sound absorption of around 0.0008 dB km−1 (0.4 dB at 500 km).

Figure 4 Absorption difference between (2018 to 2022) and (2094 to 2098) for SSP5-8.5.

Maps of the difference in 5 years mean of sound absorption (α) in dB km−1 in (2094 to 2098) and (2018 to 2022), calculated using van Moll, Ainslie & van Vossen (2009) algorithm at (A) 5, (B) 125, (C) 300 and (D) 640 m depth calculated for SSP5-8.5. The black dots indicate the sound source locations.

Figure 5 Absorption difference between (2018 to 2022) and (2094 to 2098) for SSP2-4.5.

Maps of the difference in 5 years mean of sound absorption (α) in dB km−1 in (2094 to 2098) and (2018 to 2022), calculated using van Moll, Ainslie & van Vossen (2009) algorithm at (A) 5, (B) 125, (C) 300 and (D) 640 m depth calculated for SSP2-4.5. The black dots indicate the sound source locations.

The absorption values at the selected locations (Fig. 6) show a consistent decrease in absorption at every location with a maximum decrease in the Norwegian Sea (72° N, 1° W) of 0.0019 dB km−1 (0.95 at 500 km) for SSP5-8.5 and 0.0013 dB km−1 (0.65 dB at 500 km) for SSP2-4.5.

Figure 6 Absorption profiles for the selected locations.

Sound absorption (α) in dB km−1 profiles over depth for the winter (dashed lines) and summer season (continuous lines) where in blue is boreal summer (2018 to 2022), in red winter (2018 to 2022), in yellow summer (2094 to 2098) and in purple winter (2094 to 2098) for SSP5-8.5 and green and azure for summer and winter (2094 to 2098) for SSP2-4.5 for (A) Northwest Atlantic Ocean (45° N 40° W), (B) Northeast Atlantic Ocean (47° N 14° W), (C) Norwegian Sea (72° N 1° W), (D) Arctic Ocean (75° N 140° W), (E) North Pacific Ocean (50° N 167° E), and (F) Southern Ocean (60° S 25° E).

Expected changes in sound propagation

The sound speed profile changes will lead to changes in received SPL at the different locations. The largest changes will be heard in the North Atlantic Ocean where in 2098 ship noise will travel in the sub-surface duct making the top 200 m noisier than today, that for scenario SSP5-8.5 will lead to an increase larger than 50 dB when propagating over 500 km. In the Northwest Atlantic Ocean (45° N, 40° W) today, sound propagates mainly after reflection by the sea-bottom in a weak surface duct, particularly during winter (Fig. 7). During summer, the PL due to the interaction with the sea-bottom, does not allow sound to propagate over large distances and at the surface, it mostly is heard at the convergence zones. Here the sound rays propagating in the sub-surface duct interact with the rays reflected by the sea-bottom. In contrast we observe that by the year 2098 a new sub-surface duct will allow sound in the top 200 m to propagate over large distances (>500 km). This new duct is a robust observation, observed in different model simulations, albeit that it is similar for the SSP2-4.5 scenario in which some rays will still be reflected by the sea-bottom, forming convergence zones every 45 km. In particular, at the depth of the sound speed minimum (50 m below surface) the SPLtot will be louder with increases larger than 50 dB (Fig. 8).

Figure 7 Predicted SPLtot for the Northwest Atlantic Ocean (45° N 40° W) from a single bulker and wind.

Predicted SPLtot for the Northwest Atlantic Ocean (45° N 40° W) from a single bulker and wind where (A) is winter (2018 to 2022), (B) winter (2094 to 2099) for SSP2-4.5, (C) winter (2094 to 2098) for SSP5-8.5 (D) summer (2018 to 2022), (E) summer (2094 to 2098) for SSP2-4.5 and (F) summer (2094 to 2098) for SSP5-8.5.

Figure 8 The sound pressure level from the bulker and wind.

The figure shows for the Northwest Atlantic Ocean (45° N 40° W) the SPL from the bulker (SPLship) and from the wind (SPLwind) at a single depth (50 m) for (2018 to 2022) (red), (2094 to 2098) for SSP2-4.5 (red) and SSP5-8.5 (yellow). The SPLship was calculated as the difference between the source level (SL) and the propagation loss (PL) derived using a parabolic equation model. The plot shows the SPLwind in black for (2018 to 2022) (continuous line), (2094 to 2099) SSP2-4.5 (dotted line) and SSP5-8.5 (dash-dotted line).

The SPLtot for the Northeast Atlantic Ocean (47° N, 14° W) shows a similar sound propagation (Fig. 9) to that in the Northwest Atlantic Ocean (Fig. 7). In 2098, the convergence zones will still be present, but in the winter sound will also propagate in a sub-surface duct (0 to 200 m) over >500 km for SSP5-8.5 and SSP2-4.5.

Figure 9 Predicted SPLtot for the Northeast Atlantic Ocean (47° N 14° W) from a single bulker and wind.

Predicted SPLtot for the Northeast Atlantic Ocean (47° N 14° W) from a single bulker and wind where (A) is winter (2018 to 2022), (B) winter (2094 to 2098) for SSP2-4.5, (C) winter (2094 to 2098) for SSP5-8.5 (D) summer (2018 to 2022), (E) summer (2094 to 2098) for SSP2-4.5 and (F) summer (2094 to 2098) for SSP5-8.5.

In 2098 the Norwegian Sea (72° N, 1° W, Fig. 10) will also have a new sub-surface sound duct. In the SSP2-4.5 and SSP5-8.5 scenarios, sound will propagate via the sub-surface duct, with some rays leaving the duct interacting with the sea-bottom. Today only part of the rays propagate via the surface duct and a large part of the rays reach the sea-bottom.

Figure 10 Predicted SPLtot for the Norwegian Sea (72° N 1° W) from a single bulker and wind.

Predicted SPLtot for the Norwegian Sea (72° N 1° W) from a single bulker and wind where (A) is winter (2018 to 2022), (B) winter (2094 to 2099) for SSP2-4.5, (C) winter (2094 to 2098) for SSP5-8.5 (D) summer (2018 to 2022), (E) summer (2094 to 2098) for SSP2-4.5 and (F) summer (2094 to 2098) for SSP5-8.5.

The other locations studied, such as the North Pacific Ocean (50° N, 167° E, Fig. 11), with the same SPLship will generally become quieter by 2098. In 2022 sound propagates reaching large distances (>500 km), while in 2098 at the surface the sound will be confined to the convergence zones. In the Arctic Ocean (75° N 140° W, Fig. 12) the SPLtot in 2098 will be similar to 2022, with sound propagating through the entire water column. Despite the increase of sound speed at the surface in the Southern Ocean (Fig. 13) SPLtot is expected to remain similar to today.

Figure 11 Predicted sound pressure level for the North Pacific Ocean (50° N 167° E) from a single bulker and wind.

Predicted SPLtot for the North Pacific Ocean (50° N 167° E) from a single bulker and wind where (A) is winter (2018 to 2022), (B) winter (2094 to 2098) for SSP2-4.5, (C) winter (2094 to 2098) for SSP5-8.5 (D) summer (2018 to 2022), (E) summer (2094 to 2098) for SSP2-4.5 and (F) summer (2094 to 2098) for SSP5-8.5.

Figure 12 Predicted sound pressure level for the Arctic Ocean (75° N 140° W) from a single bulker and wind.

Predicted SPLtot for the Arctic Ocean (75° N 140° W) from a single bulker and wind where (A) is winter (2018 to 2022), (B) winter (2094 to 2098) for SSP2-4.5, (C) winter (2094 to 2098) for SSP5-8.5 (D) summer (2018 to 2022), (E) summer (2094 to 2098) for SSP2-4.5 and (F) summer (2094 to 2098) for SSP5-8.5.

Figure 13 Predicted sound pressure level for the Southern Ocean (60° S 25° E) from a single bulker and wind.

Predicted SPLtot for the Southern Ocean (60° S 25° E) from a single bulker and wind where (A) is winter (2018 to 2022), (B) winter (2094 to 2099) for SSP2-4.5, (C) winter (2094 to 2098) for SSP5-8.5 (D) summer (2018 to 2022), (E) summer (2094 to 2098) for SSP2-4.5 and (F) summer (2094 to 2098) for SSP5-8.5.

In particular, for SSP5-8.5 and SSP2-4.5 the contribution of absorption (i.e., related to ocean acidification) to the future changes in SPLtot is negligible (<1 dB), with a maximum contribution of only 0.8 dB at 500 km in the Arctic Ocean.

Discussion

Global drivers of the future sound propagation

Our results show that the predicted climate change for 2098 results in an overall increase of sound speed in the top 125 m of the water column, except for parts of the North Atlantic Ocean, Labrador and Norwegian Seas. These results are consistent with Affatati, Scaini & Salon (2022) using the CESM version 1 Large Ensemble project (LENS, (Kay et al., 2015)) forced with the RCP8.5 climate change scenario, who found an increase in sound speed from 2006–2016 to 2090–2100 up to 24 m s−1 (1.5 %) in the polar regions. Consistent with our study, the only regions where they identified a decrease in sound speed, with a maximum of 10 m s−1, were parts of the Labrador Sea and North Atlantic Ocean. However, in our study for both the climate scenarios (SSP2-4.5 and SSP5-8.5) the decrease was much larger with a maximum of 20 m s−1 for SSP2-4.5 in the North Atlantic Ocean. This difference is probably partly caused by the difference in temperature and salinity projections between CESM1 and CESM2 and that in this we used 5 years when Affatati, Scaini & Salon (2022) used a 10 years mean.

At the frequency considered here (125 Hz), results show that the changes in PL are largely driven by stratification rather than sound absorption. Despite the large decrease in absorption, that in some cases is >60%, the final contribution of absorption to SPLtot is negligible 1.5 dB at 500 km. At higher frequencies, the contribution of absorption could be more important. For example, at 500 Hz absorption will decrease between 0.007 and 0.019 dB km−1, with a maximum decrease in the Norwegian Sea between 0.019 and 0.013 dB km−1 and in the North Atlantic between 0.023 and 0.082 dB km−1 at 3 kHz. In particular, previous studies (Duda, 2017; Joseph & Chiu, 2010; Reeder & Chiu, 2010; Udovydchenkov et al., 2010) showed that in some scenarios, changes in absorption can significantly alter sound propagation. For example: when sound is trapped in a duct where it propagates without interactions with the sea-surface (i.e., a deep sound channel). Duda (2017) found that in the Beaufort Sea Pacific Water duct pH will decrease by 0.2 (from 8.1 to 7.9) within the next 30–50 years and a source of 900 Hz located in the duct will thus have a SPL of 7 dB higher and sound will consequently travel 38% further. However, other studies with a surface sound source found similar results to our study, with an absolute change in SPL smaller than 2 dB (Duda, 2017; Joseph & Chiu, 2010; Reeder & Chiu, 2010; Udovydchenkov et al., 2010).

The changes in SPLtot are mainly visible in the top 200 m in the North Atlantic Ocean and Norwegian Sea, where the sub-surface duct (0 to 200 m) will become more marked. These changes are probably caused in both regions by changes in temperature and salinity profiles (Figs. S1 and S2). In particular, the decrease of surface temperature will increase the proportion of radiated power trapped in the ocean contributing to the increase of the future SPLtot. The opposite effect will characterize regions where temperatures are projected to increase (e.g., Pacific and Southern Ocean), making the surface layer quieter. Ainslie (2011) quantified this contribution in a reduction of the noise level by 8% for a temperature increase of 0.1 °C.

Effect of the AMOC on the North Atlantic Ocean sound propagation

We observed the largest changes in SPLtot in the North Atlantic Ocean and the Norwegian Sea. In these regions, stratification and consequently the sound speed profiles are controlled by the AMOC (Ivanovic et al., 2018; Haskins et al., 2020; Jackson et al., 2020). The AMOC started slowing down in the middle of the late 20th century and is still projected to continue slowing down in the next decades (Boers, 2021; Bryden, Longworth & Cunningham, 2005; Delworth & Dixon, 2000; Lynch-Stieglitz, 2017; Visbeck et al., 2001). This slowdown has been observed by direct measurements at the Rapid Climate Change array at 26.5° N (Smeed et al., 2018), from 2014 by the OSNAP observing system at higher latitudes (Susan Lozier et al., 2017) and by temperature-based and geochemical proxy reconstructions (Rahmstorf et al., 2015; Caesar et al., 2018; Thornalley et al., 2018). For the next decades, the Sixth Assessment of the United Nations Intergovernmental Panel on Climate Change (IPCC, Geneva, Switzerland) projected that in the 21st century this slowing down will continue (Skea, Shukla & Kılkış, 2022). Previous studies showed that the AMOC changes are insensitive to the climate scenario (Weijer et al., 2020), which is also the case for our study (Fig. 14). The similar decrease of AMOC strength for the two climate scenarios lead to similar correlations with the North Atlantic Ocean surface sound duct SPLtot (Fig. 15). In fact the two variables show a strong negative correlation for both climate change scenarios used, with an R2 of 0.8 for SSP5-8.5 and 0.87 for SSP2-4.5. SPLtot in the duct will increase constantly over time with a maximum increase of SPLtot of 7 dB at the end of this century. The strong observed correlation between AMOC and sound propagation shows that the AMOC will be the main driver affecting the future SPLtot in the North Atlantic Ocean. Also, the consistency in the trend for both SSP2-4.5 and SSP5-8.5 shows that the changes in climate variables are proportional to the cumulative carbon emissions (Herrington & Zickfeld, 2014; Notz & Stroeve, 2016; Steinacher & Joos, 2016), implying that the AMOC in the first decades of any SSP scenario is mostly determined by historical CO2 emissions. The mechanism behind the observed AMOC slowing is the melting of ice and changes in the hydrological cycle (Liu et al., 2020) with the consequence that surface water in the North Atlantic Ocean (>40° N) will become less saline and colder, hindering the sinking of high-density (more saline) surface water. To confirm the correlation between AMOC and the changes in sound propagation, Figs. S3 and S4 shows that the regions of the North Atlantic Ocean where sound speed decrease the seawater temperature also decrease. This weakened AMOC will slow down even more the future Arctic due to sea-ice loss, with less heat reaching the Arctic Ocean (Liu et al., 2020; Boers, 2021) which modifies the North Atlantic Ocean and Norwegian Sea seawater temperature and salinity profiles (Figs. S1 and S2). In the North Atlantic Ocean and Norwegian Sea the surface salinity will decrease and temperature will increase with the exception of the Northeast Atlantic Ocean where surface winter temperature will decrease. Other consequences are changes in the intensity and frequency of winter storms over Europe (Woollings et al., 2012) and in sea level (Pardaens, Gregory & Lowe, 2011). Therefore, understanding the changes of AMOC is key for predicting the local impact of climate change with important consequences for the society, marine life and industry. Our results are consistent with previous studies, for a RCP8.5 a scenario similar to SSP5-8.5. Liu et al. (2020) predicted by 2100 a temperature decrease up to 1.8 K between 48 to 60° N in the entire water column with the largest decrease at the surface. In our study, this change in the temperature profile lead to the formation of a new sub-surface sound duct at 150 m and a deepening of the sound channel.

Figure 14 Changes in Atlantic Meridional Overturning Circulation (AMOC) strength over time for SSP5-8.5 and SSP2-4.5.

Volumetric flow rate in Sverdrup of the Atlantic Meridional Overturning Circulation (AMOC) derived in the CESM2 for two different climate scenarios for SSP5-8.5 (in blue) and SSP2-4.5 (in red) at 26.5° N. The model data have been smoothed using a 5-year moving median.

Figure 15 Correlation between Atlantic Meridional Overturning Circulation (AMOC) strength and the sound pressure level in the top 200 m.

Volumetric flow rate in Sverdrup of the Atlantic Meridional Overturning Circulation (AMOC) derived in the CESM2 using CMIP6 climate models for two different climate scenarios SSP5-8.5 (in blue) and SSP2-4.5 (in red) at 26.5° N smoothed using a 5-year moving median vs top 200 m median of the predicted SPLtot between 475 and 500 km distance for the Northwest Atlantic Ocean (45° N 40° W) from a single bulker. The regression equations are: for SSP5-8.5: SPLtot (dB re 1 μPa) = −1.2 ± 0.1AMOC + 85.7 ± 0.8 (R2 = 0.8) and for SSP2-4.5: SPLtot (dB re 1 μPa) = −1.2 ± 0.1AMOC + 87.9 ± 0.8 (R2 = 0.87).

Likely the projected changes in sound propagation will affect maritime users that rely on sound. For example, navies have been concerned with climate change effects (National Research Council, 2010). Particularly when relying heavily on acoustic sensors and systems, their performance in the North Atlantic Ocean and the Norwegian Sea will likely be severely affected. Also, we think these changes will impact fauna, although the exact extent remains to be investigated. Likely most harm will be done to marine mammals, compromising hearing ability and inducing physiological and behavioral changes putting animals under stress. Hence, further modelling and fieldwork studies are necessary to accurately quantify these changes and fully elucidate the mechanisms behind the changes in sound propagation.

Conclusions and future studies

The strong correlation we observe between modeled sound propagation and the predicted changes in AMOC suggest that this could provide an additional toolbox to monitor AMOC changes. At the moment the AMOC is measured using transport mooring arrays equipped with dynamic height and current meters (Smeed et al., 2018; Lozier et al., 2017; Meinen et al., 2013) and indirect measurements such as satellite altimetry coupled with in situ measurements (e.g., Argo floats) (McCarthy et al., 2020). Unfortunately, there are no long observational records (or quantitative paleo proxies) of AMOC and that leads to a large uncertainty in AMOC projections with the consequence that it will take several decades to detect a forced trend in the AMOC due to the influence of internal variability (Baehr et al., 2007; Roberts & Palmer, 2012; Roberts, Jackson & McNeall, 2014). Due to this uncertainty, the Fifth report of the Intergovernmental Panel on Climate Change (IPCC, GE, Switzerland) concluded that a weakening of AMOC is likely for all scenarios, but the predicted weakening ranges between 34% and 45% (Weijer et al., 2020). The current methods along with new acoustic measurements could help to improve these predictions, with the acoustic measurements being suitable for real time monitoring. Such acoustic monitoring could be carried out using the existing (Stanistreet et al., 2017; Davis et al., 2017; Durette-Morin et al., 2019; Soldevilla et al., 2014) and new passive acoustic measurements and help to directly link the strength of AMOC with its acoustic impact on the marine ecosystem. We also suggest to use a series of surface artificial sound sources placed in different locations in the North Atlantic Ocean and measure the PL with a series of acoustic buoys. These acoustic buoys need to be equipped with several hydrophones located at different depths to capture the formation of new ducts. Combining models for climate and sound propagation we showed that climate change will significantly change the propagation of ship noise, especially the north Atlantic Ocean and the Norwegian Sea. These results are consistent between high (SSP5-8.5) and lower emission scenarios (SSP2-4.5). This implies that in the next century not only enhanced marine traffic will potentially make the future oceans noisier, but also a change in sound propagation. In the most affected regions changes will make the top 200 m noisier up to 7 dB, with possible adverse effects on marine life and maritime users that rely on sound. In other regions, the propagation of ship noise will be similar to today with some regions that will be slightly quieter (e.g., Pacific and Southern Oceans).

The most likely mechanism behind the observed change in the propagation of ship noise in the North Atlantic is a slowing down of the AMOC, which will change the sound speed profile creating a stronger sub-surface duct at 150 m. This new duct will allow ship noise to propagate over large distances (>500 km). A strong correlation is observed between SPLtot and AMOC (R2 = 0.8 for SSP5-8.5 and R2 = 0.87 for SSP2-4.5), which might open the way for future studies to quantify AMOC changes using sound propagation.

Supplemental Information

Supplemental Information 1 Temperature profiles for the selected locations.

Seawater temperature (T) in ° C profiles over depth for the winter (dashed lines) and summer season (continuous lines) where in blue is boreal summer (2018 to 2022), in red winter (2018 to 2022), in yellow summer (2094 to 2098) and in purple winter (2094 to 2098) for SSP5-8.5 and green and azure for summer and winter (2094 to 2098) for SSP2-4.5 for a) Northwest Atlantic Ocean (45° N 40° W), b) Northeast Atlantic Ocean (47° N 14° W), c) Norwegian Sea (72° N 1° W), d) Arctic Ocean (75° N 140° W), e) North Pacific Ocean (50° N 167° E) and f) Southern Ocean (60° S 25° E).

Click here for additional data file.

Supplemental Information 2 Salinity profiles for the selected locations.

Salinity (S) profiles over depth for the winter (dashed lines) and summer season (continuous lines) where in blue is boreal summer (2018 to 2022), in red winter (2018 to 2022), in yellow summer (2094 to 2098) and in purple winter (2094 to 2098) for SSP5-8.5 and green and azure for summer and winter (2094 to 2098) for SSP2-4.5 for a) Northwest Atlantic Ocean (45° N 40° W), (b) Northeast Atlantic Ocean (47° N 14° W), (c) Norwegian Sea (72° N 1° W), (d) Arctic Ocean (75° N 140° W), (e) North Pacific Ocean (50° N 167° E) and (f) Southern Ocean (60° S 25° E).

Click here for additional data file.

Supplemental Information 3 Temperature difference between (2018 to 2022) and (2094 to 2098) for SSP5-8.5.

Maps of the difference in 5 years mean of seawater temperature (T) in ° C between (2018 to 2022) and (2094 to 2098) at (a) 5 m, (b) 125, (c) 300 and (d) 640 m depth calculated for SSP5-8.5. The black dots indicate the sound source locations.

Click here for additional data file.

Supplemental Information 4 Salinity difference between (2018 to 2022) and (2094 to 2098) for SSP2-4.5.

Maps of the difference in 5 years mean of seawater temperature (T) in ° C between (2018 to 2022) and (2094 to 2098) at (a) 5 m, (b) 125, (c) 300 and (d) 640 m depth calculated for SSP2-4.5. The black dots indicate the sound source locations.

Click here for additional data file.

Additional Information and Declarations

Competing Interests

Author Contributions

Data Availability

The authors declare that they have no competing interests.

Luca Possenti conceived and designed the experiments, performed the experiments, analyzed the data, prepared figures and/or tables, authored or reviewed drafts of the article, and approved the final draft.

Gert-Jan Reichart conceived and designed the experiments, authored or reviewed drafts of the article, and approved the final draft.

Lennart de Nooijer conceived and designed the experiments, authored or reviewed drafts of the article, and approved the final draft.

Frans-Peter Lam conceived and designed the experiments, authored or reviewed drafts of the article, and approved the final draft.

Christ de Jong conceived and designed the experiments, authored or reviewed drafts of the article, and approved the final draft.

Mathieu Colin conceived and designed the experiments, authored or reviewed drafts of the article, and approved the final draft.

Bas Binnerts conceived and designed the experiments, authored or reviewed drafts of the article, and approved the final draft.

Amber Boot conceived and designed the experiments, authored or reviewed drafts of the article, and approved the final draft.

Anna von der Heydt conceived and designed the experiments, authored or reviewed drafts of the article, and approved the final draft.

The following information was supplied regarding data availability:

The project data is available at WCRP CMIP6 using the single experiment run r11i1p1f1 for ssp5-8.5 and ssp2-4.5: https://esgf-node.llnl.gov/search/cmip6/.

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
