# Peer review of "Predicting the contribution of climate change on North Atlantic underwater sound propagation"

_PeerJ, doi:10.7717/peerj.16208_

## Round 0.1 · original submission · Minor Revisions

This manuscript has been reviewed by two experts in this research field and both of them recommended a minor revision. I agree with their recommendations, as well as suggestions on how to improve the quality of this manuscript by addressing their comments.

Reviewer 1 ·

Basic reporting

The goal of this paper is to find a global effect of climate change for 2022 and 2099
using values from atmospheric and seawater temperature, salinity, pH, and wind speed.
The authors describe the methods used in detail. The study is interesting and well described and tackles an important link between underwater acoustics and climate change.


Minor English adjustments should be performed. Note the spelling seems to be following British English.


Abstract
Move “Using numerical modelling, we here explore the impact of climate change on underwater sound propagation” to after “The world-wide eûect of climate change was explored for the winter and summer seasons using..” so to have a first introductory part and the describing what you did

Please check the punctuation throughout the whole text. Below, for the abstract:
-Since the industrial revolution,
-This is mainly caused…”This” unclear antecedent
-resource exploration, and
-an increase in seawater
-seawater temperature and
-a decrease in ocean pH.
-frequencies (<10 kHz), enhancing
-At the same time, temperature
- The worldwide effect of climate change
-was explored for the winter and summer seasons using the 2022 and 2099 (projected) atmospheric and seawater temperature, salinity, pH and wind speed as input.
-concentration-driven SSP2-4.5
-5, 125, 300, and 640 m
-North Atlantic Ocean and the Norwegian Sea, where
-This decrease in sound speed results
-two climate change scenarios, with an increase


Introduction

The first part seems to be unbalanced towards discussing acidification. I would use this part to introduce all the topics you will refer to in your manuscript.

54 Humankind has introduced more than 330 petagrams

Lines 74-76 I understand what you mean, but I suggest to reframe it better
106 soundscape and compare different scenarios






Raw data doesn’t seem to be provided. The link provided as a repository seems to be leading to the cmpi6 page https://esgf-node.llnl.gov/search/cmip6/ (if I checked it correctly).

Experimental design

Methods described with sufficient detail & information to replicate.
Although scientifically sound, I find the materials and methods section a bit cumbersome to read. I suggest finding another way to describing your methods in a different way, perhaps through a table.

Validity of the findings

Discussion:
L349-351 we think these changes will impact fauna, but we are not sure yet…

355 Potential monitoring of AMOC variability using sound propagation?
This is not very clear to me.

I find this part very interesting; however, I would mention potential challenges related to what you suggest (e.g., cost of long-term acoustic measurements).

Conclusions
Also, I find the previous chapter being an evolution of the discussion, not just a discussion of your results. I suggest to include it into the conclusions, maybe changing the chapter name to “conclusions and future studies”.

Additional comments

Title: you mention the impact. However, you imply the impact that will derive from the results of your work, but you do not actually compute impacts. I advise changing the title to reflect the results of your manuscript.

Please be careful when using the concept of “climate” and results derived from 1-year data, since climatic change are usually computer over larger temporal scales. Also the AMOC varies on a multidecadal timescale

I would also make clearer at some point in the paper that 125 Hz is one of the frequency values MSFD considers, 63 Hz being the other one (plus possible higher ones e.g., 1kHz as indicated by JOMOPANS)

I would also be careful with the use of the world “soundscape”. If we define it as “the collection of biological, geophysical and anthropogenic sounds that emanate from a landscape and which vary over space and time reflecting important ecosystem processes and human activities.” Pijanowski, B. C., Farina, A., Gage, S. H., Dumyahn, S. L., and Krause, B. L. (2011). What is soundscape ecology? An introduction and overview of an emerging new science. Lands. Ecol. 26, 1213–1232. doi: 10.1007/s10980-011-9600-8



Good luck with your reviews

Reviewer 2 ·

Basic reporting

see pdf file

Experimental design

see pdf file

Validity of the findings

see pdf file

Additional comments

see pdf file

Annotated reviews are not available for download in order to protect the identity of reviewers who chose to remain anonymous.

---

## Round 0.2 · accepted · Accept

I believe the authors have addressed all review comments so I am happy to accept it. Congratulations! I hope you will submit more manuscripts to our journal in the future.